# TransBoost: Improving the Best ImageNet Performance using Deep Transduction

**Omer Belhasin**[*]
Department of Computer Science
Technion - Israel Institute of Technology
omer.be@cs.technion.ac.il

**Guy Bar-Shalom**[*]
Department of Computer Science
Technion - Israel Institute of Technology
guy.b@cs.technion.ac.il

**Ran El-Yaniv**
Department of Computer Science
Technion - Israel Institute of Technology, Deci.AI
rani@cs.technion.ac.il

## Abstract

This paper deals with *deep transductive learning*, and proposes *TransBoost* as a procedure for fine-tuning any deep neural model to improve its performance on any (unlabeled) test set provided at training time. TransBoost is inspired by a large margin principle and is efficient and simple to use. Our method significantly improves the ImageNet classification performance on a wide range of architectures, such as ResNets, MobileNetV3-L, EfficientNetB0, ViT-S, and ConvNext-T, leading to state-of-the-art transductive performance. Additionally we show that TransBoost is effective on a wide variety of image classification datasets. The implementation of TransBoost is provided at: https://github.com/omerb01/TransBoost.

## 1  Introduction

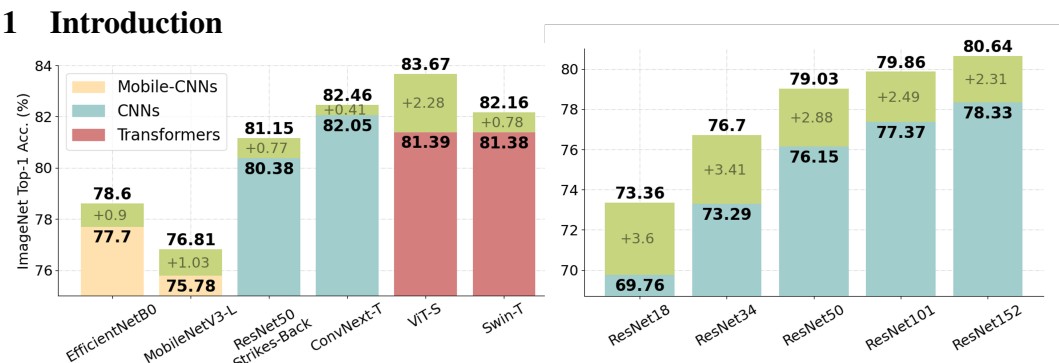

Figure 1: ImageNet top-1 accuracy gains of TransBoost (green) using representatives of CNNs (blue), Mobile-CNNs (yellow) and Transformers (red), in comparison to the standard inductive (fully supervised) performance.

Alternative learning frameworks that make use of unlabeled data have gained considerable attention in recent years. These include *semi-supervised learning* and *transductive learning*, both of which were extensively studied in classical and statistical machine learning [1, 2, 3] before the advent of deep learning. In this article we focus on deep *transductive* classification. In this setting, both a labeled training sample and an (unlabeled) test sample are provided at training time. The goal is to predict *only* the labels of the given test instances as accurately as possible. In contrast, in standard

---

[*]Equal contribution.

36th Conference on Neural Information Processing Systems (NeurIPS 2022).

*inductive* classification, the goal is to train a general model capable of predicting the labels of unseen test instances.

In semi-supervised learning (SSL) we are also given a set of unlabeled examples, which is used for selecting a model. Despite a commonly held misconception, however, the goal of SSL remains inductive rather than transductive, and the trained model must predict labels for new instances (i.e., the unlabeled instances are used for training only).

Transductive models deliver higher accuracy gains compared with traditional inductive models, as we demonstrate in this paper in the context of image classification. Much of the power of transductive learning is achieved through analyzing the given test set as a group (see Section 5.4). Transductive prediction is thus especially useful whenever we can accumulate a test set of instances and then train a specialized model to predict their labels. While there are many possible relevant use cases, one that stands out is medical diagnosis. Here, daily or weekly medical records can be accumulated and sent to the transductive predictor as a set. In Appendix A, we highlight a number of additional use cases.

Perhaps the best-known transductive classification method in classical machine learning is the transductive support vector machine (TSVM) [4, 5]. While a support vector machine (SVM) [6] seeks to achieve a general decision function, a TSVM attempts to reduce misclassifications of just the target test instances. To the best of our knowledge, in deep learning, transductive settings have only been addressed in the case of few-shot learning (t-FSL).

In this paper we address deep transductive learning and introduce *TransBoost*, a novel procedure that takes a pretrained neural classification model, along with a (labeled) training set, and an (unlabeled) test set. The procedure fine-tunes the model to improve its performance on this particular test set. The TransBoost optimization procedure is inspired by TSVM and its objective attempts to approximate a *large margin principle*. TransBoost is implemented through a simple transductive loss component, which is combined with the standard cross-entropy loss.

We examine the effectiveness of TransBoost by using a variety of pretrained neural networks, datasets, and baseline models. Our extensive study examines a spectrum of ResNet architectures and a variety of modern architectures such as ConvNext [7], MobileNetv3 [8] and ViT [9]. Figure 1 highlights some of TransBoost's results on the ImageNet dataset, which demonstrate impressive and consistent accuracy gains, leading to state-of-the-art performance compared to standard (inductive) classification as well as versus SSL and t-FSL methods (see full results in Section 5.2).

To summarize, the contributions of this paper are: (1) A novel deep transductive loss function, inspired by a large margin principle, which optimizes performance on a specific test set. (2) A simple and efficient transductive procedure for fine-tuning a pretrained neural model in order to boost its performance over an inductive setting. (3) A comprehensive empirical study of the proposed fine-tuning procedure showing great benefits across a large number of architectures, datasets and baseline techniques.

## 2 Problem Formulation

As transductive learning has not been extensively researched in the deep learning community, we begin by defining the transductive learning problem within the context of deep learning classification. We follow Vapnik's standard formulation from statistical learning theory [4]. Let $P(X, Y)$ be a probability distribution over $\mathcal{X} \times \mathcal{Y}$, where $\mathcal{X}$ represents an input space (e.g., raw image data), and $\mathcal{Y}$ represents a label set corresponding to $C$ classes. The learner is provided with a set of labeled instances, $S_l \triangleq \{(x_1, y_1), \ldots, (x_L, y_L)\}$, where $x_i \in \mathcal{X}$, $y_i \in \mathcal{Y}$, and a finite set of unlabeled instances, $X_u \triangleq \{x_{L+1}, \ldots, x_{L+U}\}$, where $x_i \in \mathcal{X}$. The objective is to label the unlabeled instances based on this data. Given a deep neural network $f_\theta : \mathcal{X} \to \mathcal{Y}$, where $\theta$ denotes its parameters, and a loss function $\ell : \mathcal{Y} \times \mathcal{Y} \to \mathbb{R}$, we follow [1] (Setting 2) and define the (true) *risk* over the unlabeled set,

$$E_{f_\theta}(X_U) \triangleq \frac{1}{U} \sum_{i=1}^{U} \ell(f_\theta(x_{L+i}), y_{L+i}). \tag{1}$$

Thus, we are given a labeled training set $S_l \triangleq (X_l, Y_l)$ selected i.i.d. according to $P(X, Y)$, as well as a pretrained classification model $f_\theta$, and we presume it was trained with the labeled set $S_l$. An independent test set $S_u \triangleq (X_u, Y_u)$, of $U$ samples is randomly and independently selected in

the same manner, and we are required to revise the given model based on both $S_l$ and $X_u$, so as to minimize Equation (1) without knowing the labels $Y_u$.

## 3 Related Work

Transductive learning (or *transduction*) was first formulated by Vapnik [4, 10] who also introduced the first transductive algorithm – TSVM [4, 5], which learns a large margin hyperplane classifier from labeled training data while simultaneously forcing it to take into account the (unlabeled) test data. Whereas transduction received considerable attention in the context of classical machine learning [1, 2, 3], it has only been briefly addressed in the context of deep neural networks.

Recall that in (deep) transductive classification we are provided with both a labeled training sample and an unlabeled test sample. A related but different setting is *semi-supervised learning* (SSL), where in addition to the labeled training set, we are also given a (typically large) unlabeled training sample. The objective is *inductive* – namely, to guess the label of any *unseen* test sample (that was drawn from the same distribution). A simple observation is that any SSL algorithm can be applied in transduction by using the given (unlabeled) test sample as the unlabeled SSL training sample. Thus, algorithms such as [11, 12, 13] can be used to solve transductive learning. This fact may explain a common confusion between SSL and transduction. Nevertheless, an inductive SSL algorithm does not need to use (and cannot rely on) the fact that it will only be asked about the given test points. In contrast, a (meaningful) transductive algorithm must rely on this knowledge, as we contend in this paper. In fact, in Section 5.2 we demonstrate that our proposed transductive procedure substantially outperforms the *SimCLR* [11] and *SimCLRv2* [12] algorithms, where SimCLRv2 currently achieves state-of-the-art SSL performance according to [14].

Recently, there has been a flurry of research concerning *transductive few-shot learning* (t-FSL) [15, 16, 17]. In t-FSL we are given a classifier for a certain classification task, such as cats versus dogs, as well as a training set (called the *support* set) for unseen classes (referred to as *ways*), such as elephants, tigers and lizards, containing only few labeled samples per unseen class. A test set (called the *query* set), containing instances from the unseen classes, is given and the goal is to learn the additional classes so as to correctly classify these test instances. Clearly, this setting is transductive; however, the goal here is to learn from a few labeled examples (e.g., one-shot, five-shot). Formally, a t-FSL algorithm can be applied to transductive classification (by treating the training set as a support set). For the most part, however, methods for t-FSL cannot be effectively applied to large (non few-shot) training sets (e.g., iLPC [17]). Nevertheless, in order to conduct a complete study in transductive classification, we chose two t-FSL representative algorithms as baselines for our experiments (see Section 5). Specifically, we consider the entropy minimization (Ent-min) method of [15], and the TIM-GD method of [16]. With Ent-min, the Shannon entropy is reduced by fine-tuning, whereas TIM-GD optimizes the mutual information of test instances using only the classifier weights with fine-tuning. Ent-min is a common t-FSL baseline while TIM-GD is a top performing 5-shot t-FSL algorithm according to the few-shot leaderboard [18].

## 4 TransBoost: Fine-Tuning via Transductive Learning

We now present *TransBoost*, a procedure that takes a pretrained neural model along with its (labeled) training set as well as an (unlabeled) test set. The procedure then fine-tunes the model to improve its performance for this specific test set. The TransBoost optimization objective, $\mathcal{L}$, is driven by Equation (3) and approximated by Algorithm 1, which includes a novel transductive loss component in addition to the standard cross-entropy loss. Our transductive component, $\mathcal{L}_{\text{TransBoost}}$, is introduced in Equation (3) as a regularization term and is defined in Equation (2). $\mathcal{L}_{\text{TransBoost}}$ is inspired by TSVMs [4, 5] and follows a *large margin principle*.

Let us elaborate on our loss function, $\mathcal{L}_{\text{TransBoost}}$ (2), which is applied on the set of test samples, $X_u$. This loss function includes an unsupervised non-negative symmetric pairwise similarity function, $\mathcal{S} : \mathcal{X} \times \mathcal{X} \to \mathbb{R}$, which measures the similarity of two input instances. In our implementation, we simply used an $L_2$-based score function applied on prediction (softmax) vectors generated by $f_\theta$; see Equation (4). $\mathcal{S}$ is only applied on pairs of test instances that are likely to belong to different classes. The likelihood is determined by using an implementation of the Kronecker delta function, $\delta : \mathcal{X} \times \mathcal{X} \to \{0, 1\}$, which obtains 1 iff the instances are predicted to belong to different classes;

---

**Algorithm 1:** TransBoost Procedure

---

**Input :** Labeled train sample $S_l \triangleq (X_l, Y_l)$, unlabeled test sample $X_u$, a model $f_\theta$ that was pretrained on $S_l$, a supervised pointwise loss function $\ell$, implementations of TransBoost's functions: $\mathcal{S}$, $\delta$ and $\kappa$. Hyperparameters: Number of epochs $E$, labeled batch size $L'$, unlabeled batch size $U'$, regularization parameter $\lambda$.

1 **for** $epoch \leftarrow 1$ **to** $E$ **do**
2    // Sample labeled and unlabeled batches in a cyclical manner
3    **for** $S'_l \triangleq (X'_l, Y'_l) \subset S_l$ , $X'_u \subset X_u$ **to** $\max\{\lceil \frac{L}{L'} \rceil, \lceil \frac{U}{U'} \rceil\}$ **do**
4       // Prepare random test sample pairs
5       ${X'_u}^\pi \leftarrow$ generate a random permutation of $X'_u$
6       // Approximate the TransBoost loss
7       $L_{\text{TransBoost}} \leftarrow 0$
8       **for** $i \leftarrow 0$ **to** $U'$ **do**
9          $L_{\text{TransBoost}} \leftarrow L_{\text{TransBoost}} + \kappa(x_i)\kappa({x_i}^\pi)\delta(x_i, {x_i}^\pi)\mathcal{S}(x_i, {x_i}^\pi)$
10       **end for**
11       $L_{\text{TransBoost}} \leftarrow L_{\text{TransBoost}} / \sum_{i=1}^{U'} \delta(x_i, {x_i}^\pi)$ if $\sum_{i=1}^{U'} \delta(x_i, {x_i}^\pi) \neq 0$ else $0$
12       // Apply an optimization step
13       $\theta \leftarrow$ optimize $\left[ \frac{1}{L'} \sum_{i=1}^{L'} \ell(f_\theta(x_i), y_i) + \lambda \cdot L_{\text{TransBoost}} \right]$
14    **end for**
15 **end for**

---

see Equation (5) for our implementation. Additionally, for each pair we intensify the loss using the model's confidence in its predictions. We define $\kappa : \mathcal{X} \to \mathbb{R}^+$ to be a confidence function, which for each instance, $x$, gives its class prediction confidence; see Equation (6) for our implementation. Finally, our transductive loss function, $\mathcal{L}_{\text{TransBoost}}$, is

$$\mathcal{L}_{\text{TransBoost}}(X_u | f_\theta, \mathcal{S}, \delta, \kappa) \triangleq \frac{1}{U_\delta} \sum_{1 \leq i < j \leq U} \kappa(x_i)\kappa(x_j)\delta(x_i, x_j)\mathcal{S}(x_i, x_j), \tag{2}$$

where $U_\delta \triangleq \sum_{1 \leq i < j \leq U} \delta(x_i, x_j)$. The final loss function of our fine-tuning procedure includes the transductive loss term (2) as a regularization term as follows,

$$\mathcal{L}(X_l, Y_l, X_u | f_\theta, \mathcal{S}, \delta, \kappa) \triangleq \underbrace{\frac{1}{L} \sum_{i=1}^{L} \ell(f_\theta(x_i), y_i)}_{\text{labeled/inductive (standard) loss}} + \lambda \cdot \underbrace{\mathcal{L}_{\text{TransBoost}}(X_u | f_\theta, \mathcal{S}, \delta, \kappa)}_{\text{unlabeled/transductive loss}}, \tag{3}$$

where $\ell$ is a standard (inductive) pointwise loss, e.g., the cross-entropy loss and $\lambda \in \mathbb{R}$ is a regularization hyperparameter.

Using the optimization objective in Equation (3), we incentivize test sample pairs that are likely to be different in their classes to also be different in their empirical class probabilities while preserving the prior knowledge of $f_\theta$. Accordingly, given two test samples, $(x_i, x_j)$, which are predicted to be different in their classes by $\delta$, we obtain a higher cost (by $\mathcal{S}$) if $x_i$ is close (similar) to $x_j$ in its prediction vector than if $x_i$ is far (dissimilar) from $x_j$, depending on the model's confidence in its predictions utilizing $\kappa$.

The computation of the $\mathcal{L}_{\text{TransBoost}}$ loss component (2) is quadratic in $U$ – namely, its time complexity is $O(\binom{U}{2})$. For large test sets (e.g., ImageNet), this is prohibitively large. Therefore, we propose to approximate $\mathcal{L}_{\text{TransBoost}}$ by using random subsets of pairs from the test set. This random sampling is applied for each minibatch. The pseudo-code of the optimization procedure is presented in Algorithm 1. As can be seen, the size of the set of sampled test pairs used to approximate (2) is taken to be the batch size ($U'$), which allows for very fast computation of $\mathcal{L}_{\text{TransBoost}}$ but could be sub-optimal (this size was not optimized).

## 4.1 TransBoost Implementation

While $\mathcal{S}$, $\delta$ and $\kappa$ in Equation (2) can be implemented in many ways, we now describe our proposed implementation that was used to obtain all the results described in Section 5. As can be seen below,

we attempted to instantiate Equation (2) in the simplest possible manner. The consideration of more sophisticated methods is left for future work (and an alternative choice is discussed in Section 5.5).

The unsupervised symmetric pairwise similarity function $\mathcal{S}_f$ was straightforwardly implemented using the $L_2$ norm as follows,

$$\mathcal{S}_f(x_i, x_j) \triangleq \sqrt{2} - \|\hat{p}(x_i|f_\theta) - \hat{p}(x_j|f_\theta)\|_2 \,, \tag{4}$$

where $\hat{p}(x|f_\theta)$ denotes the empirical class probability of $f_\theta$ applied on $x$. It is easy to show that $\mathcal{S}_f$ is non-negative[*] (see Appendix B for a proof).

Additionally, we implemented $\delta_f$ (5) and $\kappa_f$ (6) based on the pretrained weights of the model before it was fine-tuned with our procedure, which we refer to as $\theta_0$. Thereafter, the unsupervised symmetric pairwise selection function $\delta_f$ is implemented using pseudo-labels computed by the given pretrained model $f_{\theta_0}$,

$$\delta_f(x_i, x_j) \triangleq \begin{cases} 1 & f_{\theta_0}(x_i) \neq f_{\theta_0}(x_j) \\ 0 & \text{otherwise,} \end{cases} \tag{5}$$

and the confidence function $\kappa_f$ is implemented using the standard softmax response, i.e.,

$$\kappa_f(x) \triangleq \max\{\hat{p}_j(x|f_{\theta_0})\}_{j=1}^C, \tag{6}$$

where $\hat{p}(x|f_{\theta_0})$ is the empirical class probabilities of $f_{\theta_0}$ applied to $x$, and $C$ is the number of classes.

Finally, the optimization objective (3) is implemented using the standard cross-entropy loss, $\text{CE}(X_l, Y_l|f_\theta) \triangleq \frac{1}{L}\sum_{i=1}^L -\log \hat{p}_{y_i}(x_i|f_\theta)$, as well the proposed $\mathcal{S}_f$, $\delta_f$ and $\kappa_f$,

$$\mathcal{L}(X_l, Y_l, X_u|f_\theta, \mathcal{S}_f, \delta_f, \kappa_f) = \text{CE}(X_l, Y_l|f_\theta) + \lambda \cdot \mathcal{L}_{\text{TransBoost}}(X_u|f_\theta, \mathcal{S}_f, \delta_f, \kappa_f), \tag{7}$$

where $\lambda \in \mathbb{R}$ is a regularization hyperparameter.

## 4.2 A Large Margin Analogy of TransBoost

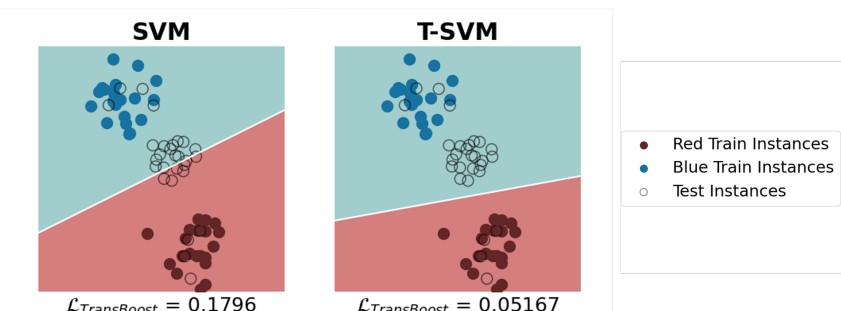

Figure 2: A toy quantitative example indicates that our $\mathcal{L}_{\text{TransBoost}}$ encourages behavior similar to TSVM.

TransBoost is inspired by TSVM, in which a large margin principle is applied to the given (unlabeled) test sample to leverage the knowledge of this specific test set. In Figure 2 we present a toy quantitative (graphical) example indicating that our transductive loss component, $\mathcal{L}_{\text{TransBoost}}$, encourages behavior similar to TSVM. Consider Figure 2 (left) showing the decision boundary obtained by SVM for the given training set (and ignoring the test points). Using SVM to implement $\delta$ (to determine if two test points are in the same class), we get that $\mathcal{L}_{\text{TransBoost}} = 0.1796$ (shown at the bottom). Clearly, this loss value is significantly larger than the corresponding loss (0.05167) obtained when using TSVM (right), where a large margin principle of test instances is applied.

# 5 Empirical Study

In this section we present a comprehensive empirical study of TransBoost. We begin by describing our experimental design and then present our studies in detail.

---

[*]$\mathcal{S}_f(x_i, x_j) = 0$ iff $\hat{p}(x_i|f_\theta)$ and $\hat{p}(x_j|f_\theta)$ are one-hot vectors where the ones indicate different classes.

## 5.1 Experimental Design and Details

This section describes the experimental details: datasets used, architectures used, and our TransBoost's procedure details and hyperparameters.

**Datasets and Preprocessing.** Most of our study of TransBoost is done using the well-known ImageNet-1k ILSVRC-2012 dataset [19], which contains 1,281,167 training instances and 50,000 test instances in 1,000 categories. Preprocessing for training is standard and includes resizing and random cropping to 224×224, followed by a random horizontal flip. For testing, the images are resized and center cropped. In some of our experiments involving advanced architectures from the Timm repository [20], we applied the test augmentations (at test time only) using its code from the Timm repository. Additionally, we investigated the effectiveness of TransBoost on the Food-101 [21], CIFAR-10 and CIFAR-100 [22], SUN-397 [23], Stanford Cars [24], FGVC Aircraft [25], the Describable Textures Dataset (DTD) [26] and Oxford 102 Flowers [27]. For each of these datasets, we applied the same preprocessing as in ImageNet, with the exception of CIFAR-10 and CIFAR-100, where we used random cropping with padding (cropping to 32×32), followed by a random horizontal flip, a single RandAugment [28] operation and MixUp [29].

**Architectures.** We examined various inductive (standard) pretrained architectures as baselines for TransBoost. Specifically, we used a variety of ResNet [30] architectures: ResNet18, ResNet34, ResNet50, ResNet101 and ResNet152, which were pretrained using the standard PyTorch training recipe [31]. We also considered ResNet50-StrikesBack [32], which is the ResNet50 architecture trained with Timm's [20] advanced training recipe. Additionally, we show results with more advanced architectures spanning various families: EfficientNetB0 [33] and MobileNetV3-L [8], ConvNext-T [7], and the Transformer models ViT-S [9] and Swin-T [34], which were pretrained with advanced procedures of Timm's library [20] in accordance to their original papers.

**Details on the TransBoost Fine-Tuning Procedure.** Throughout all TransBoost's experiments, we followed Algorithm 1, and our functions were implemented as described in Section 4.1. We used the SGD optimizer (Nesterov) with a momentum of 0.9, weight decay of $10^{-4}$, and a batch size of 1024 (consisting of 512 labeled training instances and 512 unlabeled test instances). Unless otherwise specified, the learning rate was fixed to $10^{-3}$ with no warmup for 120 epochs. The regularization hyperparameter of our loss in all our experiments was fixed to $\lambda = 2$; see Equation (7). Our hyperparameters were fine-tuned based on a validation set that was sampled from the training set over ImageNet.

## 5.2 ImageNet Experiments: Transductive vs. Inductive

This study examines the **transductive** performance of TransBoost applied to various pretrained models (all described in Section 5.1), and the transductive performance of the other baselines (SSL and t-FSL) versus the **inductive** performance of all baselines. Table 1 presents all the results. We note that some of these results have been already highlighted in Figure 1.

Table 1 is divided into several horizontal sections. In the first section we present TransBoost applied to a number of ResNet architectures. In the second section, we discuss modern architectures, including two types of Vision Transformers, and also ConvNext-T as representative of the ConvNext family, which presently achieves state-of-the-art results over ImageNet [14]. Next we present the SimCLR and SimCLRv2 SSL methods (see Appendix C for more details on how we trained them in transduction). We note that SimCLRv2 presently dominates SSL performance over ImageNet according to [14]. The last table section considers two popular t-FSL methods (which are described in Section 3). The column structure of Table 1 is straightforward but the most important column that deserves some explanation is the *Inductive / Transductive* column ($4^{th}$ column) in which we compare the transductive performance of each experiment to the relevant inductive fully supervised performance. For example, TransBoost is capable of increasing the ImageNet top-1 accuracy of ViT-S by 2.28%.

Table 1 and Figure 1 demonstrate that fine-tuning with TransBoost consistently and significantly improves the inductive top-1 accuracy performance in all cases. Even the tiny version of the top performing ConvNext architecture family (presently the state-of-the-art according to [14]) is improved by TransBoost. Surprisingly, ViT-S performed better than ConvNext-T when taking advantage of the given test samples using TransBoost, indicating that members of the ViT family could be the best transductive performers on ImageNet. A second interesting fact is that ViT-S and Swin-T

| Method | Architecture | Params (M) | Inductive / Transductive Top1 Acc. (%) | Improv. (%) |
|---|---|---|---|---|
| *ResNet [30] architectures that were pretrained using standard simple procedures:* | | | | |
| **TransBoost** | ResNet18 | 11.69 | 69.76 / **73.36** | +3.60 |
| | ResNet34 | 21.80 | 73.29 / **76.70** | +3.41 |
| | ResNet50 | 25.56 | 76.15 / **79.03** | +2.88 |
| | ResNet101 | 44.55 | 77.37 / **79.86** | +2.49 |
| | ResNet152 | 60.19 | 78.33 / **80.64** | +2.31 |
| *Advanced vision architectures that were pretrained using advanced procedures:* | | | | |
| **TransBoost** | EfficientNetB0 [33] | 05.29 | 77.70 / **78.60** | +0.90 |
| | MobileNetV3-L [8] | 05.48 | 75.78 / **76.81** | +1.03 |
| | ResNet50-StrikesBack [32] | 25.56 | 80.38 / **81.15** | +0.77 |
| | ConvNext-T [7] | 28.59 | 82.05 / **82.46** | +0.41 |
| | ViT-S [9] | 22.05 | 81.39 / **83.67** | +2.28 |
| | Swin-T [34] | 28.29 | 81.38 / **82.16** | +0.78 |
| *Baseline representatives of semi-supervised learning algorithms in transduction:* | | | | |
| SimCLR [11] | ResNet50 | 25.56 | **76.15** / 75.95 | -0.20 |
| | ResNet152 | 60.19 | **78.33** / 78.00 | -0.33 |
| SimCLRv2 [12] | ResNet50 | 25.56 | **76.15** / 74.68 | -1.47 |
| | ResNet152 | 60.19 | **78.33** / 76.64 | -1.69 |
| *Baseline representatives of transductive few-shot learning algorithms:* | | | | |
| Ent-min [15] | ResNet50 | 25.56 | 76.15 / **77.55** | +1.40 |
| TIM-GD [16] | ResNet50 | 25.56 | **76.15** / 71.51 | -4.64 |

Table 1: **Inductive vs. Transductive.** A comprehensive analysis of the transductive performance on ImageNet compared to the standard inductive (fully supervised) performance. TransBoost's performance is highlighted in green, SSL's performance is highlighted in blue, and t-FSL's performance is highlighted in yellow.

perform similarly in the inductive setting while the ViT-S top-1 accuracy gain is improved by almost x3 relative to Swin-T. Could this advantage be related to the architecture? Interestingly, TransBoost also improves the performance of ResNet50-StrikesBack, which is the standard ResNet50 architecture trained using a sophisticated training procedure [32] that boosts its top-1 accuracy by +4.2%. Surprisingly, TransBoost adds almost +1% on top of this phenomenal improvement. Another striking related result is that TransBoost improves the standard ResNet50 so that it achieves performance close to ResNet50-StrikesBack despite the fact that our procedure is much simpler than the sophisticated training procedure of [32].

Consider the first section of Table 1 (the ResNets section). While TransBoost always improves the inductive baselines, it is evident that its relative advantage monotonically decreases with the model size/performance. For example, the best transductive improvement is achieved for ResNet18 yielding a top-1 accuracy gain of +3.6% and resulting in a top-1 accuracy of 73.36%[*]. With respect to the other related baselines, TransBoost consistently outperforms both the SSL and t-FSL baseline algorithms. Comparing the results of SimCLR to the results of SimCLRv2, we see that SimCLR outperforms SimCLRv2 in the transductive setting (they mainly differ by the additional stage of knowledge distillation [36] that is added to SimCLRv2), which suggests that the stage of knowledge distillation can hinder the transductive performance. We hypothesize that distilling knowledge from test instances leads to overfitting of incorrect class predictions that lead to model confusion. Finally, when evaluating the t-FSL representatives, we observe that Ent-min improves its inductive baseline while TIM-GD (which is an upgraded version of Ent-min) is inferior to its inductive baseline.

## 5.3 TransBoost's Performance on Additional Vision Datasets

We now examine the performance of TransBoost across eight other datasets (all described in Section 5.1). Throughout this study, we start with a ResNet50 architecture pretrained on ImageNet (using PyTorch's standard training recipe). Then we apply inductive fine-tuning for each specific dataset and, finally, we apply our TransBoost procedure (see Appendix D.1 for more details).

---

[*]This ResNet18 transductive performance even outperforms the current state-of-the-art inductive performance [14] for ResNet18 on ImageNet (73.19%) [35], which is achieved with a fancy training recipe.

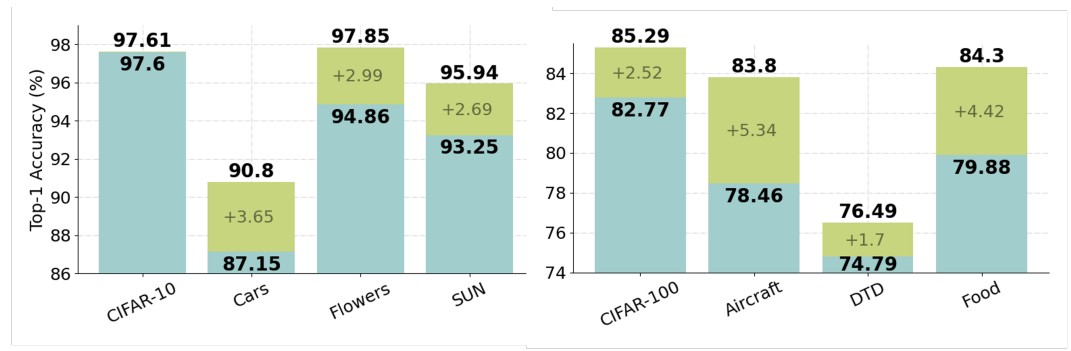

Figure 3: TransBoost's transductive performance using ResNet50 (green) on various vision datasets.

The results are visually depicted in the graph of Figure 3 and are also shown in Table S.1 in the Appendix. Clearly, TransBoost outperforms its inductive baseline in all cases. The improvement is large and significant in all datasets with the exception of CIFAR-10 where the improvement is minor. The most prominent result is obtained for the FGVC Aircraft dataset, where we observe a striking +5.34% top-1 accuracy gain from 78.46% to 83.8%. For a more in-depth discussion, we refer the reader to Appendix E.

## 5.4 Transductive Performance Improves with Test Set Size

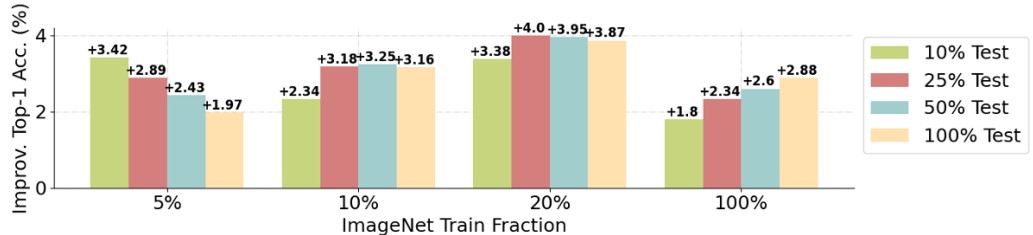

Figure 4: TransBoost's transductive performance using ResNet50 on ImageNet training and test subsets.

In this study, we consider various transductive settings, where each setting is characterized by a certain training set size and a certain test set size. We are interested in examining the final ImageNet transductive behavior of ResNet50 as a function of these sizes. The sizes of our training sets are: 5%, 10%, 20%, 100%; and for testing sets are: 10%, 25%, 50%, 100%, all taken as a fraction of the original train/validation set sizes, respectively.

Consider Figure 4 showing the top-1 accuracy gains for all combinations ($4 \times 4$ experiments in total). Importantly, we observe that whenever TransBoost uses the entire training set, its performance increases as the test set size grows. This strongly indicates that TransBoost leverages the test set as a whole, a desirable property of a transductive learning algorithm. On the other hand, and quite surprisingly, when TransBoost only used 5% of the training set, increasing the test set size worsened its performance. We hypothesize that this type of behavior is a result of poor prior knowledge (very small training set) that leads the models to wrongly analyze the relationships (similarities in our case) between pairs. This overall bad effect increases as the test set size expands. For a detailed description/discussion of this section's experiments, see Appendix E.

## 5.5 Bringing the Like Together or Separating the Difference?

TransBoost is driven by Equation (2) whose informal desideratum is: revise the model so as to take apart the representations of two instances that are likely ($\kappa$) to belong to different classes ($\delta$), which currently appear to be similar ($\mathcal{S}$). Rather than, or in addition to separating instances that are likely to be different, we could consider grouping together instances that are likely to belong to the same class. In this section we summarize our experiments to apply TransBoost for this alternative objective and for a combination of both objectives. In our study we used the ResNet50 architecture,

which was pretrained on ImageNet (using PyTorch's standard training recipe). The transductive performance of TransBoost was analyzed using three variants of the transductive loss component: (1) the proposed separation component, $\mathcal{L}_{\text{TransBoost}}(X_u|\mathcal{S}_f, \delta_f, \kappa_f)$, as in Equation (7); (2) a "reciprocal" instantiation of this objective – namely, $\mathcal{L}_{\text{TransBoost}}(X_u|\sqrt{2} - \mathcal{S}_f, 1 - \delta_f, \kappa_f)$, which brings together similar instances; and (3) a variant that considers both these objectives together. The results are presented in Table 2.

| Transductive Loss | Inductive / Transductive | |
| :---: | :---: | :---: |
| | Top1 Acc. (%) | Improv. (%) |
| $\mathcal{L}_{\text{TransBoost}}(X_u|\mathcal{S}_f, \delta_f, \kappa_f)$ | 76.15 / **79.03** | +2.88 |
| $\mathcal{L}_{\text{TransBoost}}(X_u|\sqrt{2} - \mathcal{S}_f, 1 - \delta_f, \kappa_f)$ | **76.15** / 74.64 | -1.51 |
| $\mathcal{L}_{\text{TransBoost}}(X_u|\mathcal{S}_f, \delta_f, \kappa_f) + \mathcal{L}_{\text{TransBoost}}(X_u|\sqrt{2} - \mathcal{S}_f, 1 - \delta_f, \kappa_f)$ | 76.15 / **79.00** | +2.85 |

Table 2: Performance of transductive loss variants using ResNet50. Our proposed loss is highlighted in green.

Clearly (but not significantly) the best performance is achieved using our proposed implementation (row highlighted in green). We chose our proposed loss component over the combination of both objectives because it is simpler. Interestingly, the reciprocal variation significantly degrades the top-1 performance.

# 6   Concluding Remarks

We presented TransBoost, a novel and powerful transductive fine-tuning procedure that can be efficiently applied on any pretrained model. TransBoost is a deep transductive classification algorithm that is inspired by a large margin principle such as the TSVM algorithm. Strikingly, TransBoost consistently and significantly improves the inductive ImageNet top-1 performance of many architectures including the most advanced ones, leading to state-of-the-art transductive performance. Moreover, TransBoost is effective on a broad range of image classification datasets.

As described in Algorithm 1, TransBoost is a general optimization procedure that can be implemented in a range of ways depending on the similarity function ($\mathcal{S}$), the selection function ($\delta$), and the confidence function ($\kappa$). We instantiated these functions simply and straightforwardly. A question for future research may be to ask whether more sophisticated choices will lead to better transduction performance. The ViT-S architecture achieved the best transductive classification performance in our experiments and accrued the largest gain relative to induction among advanced architectures. It would be interesting to explore the question of how this Transformer architecture facilitates this gain. In general, we can ask what is the best architecture for transductive classification. Finally, it would be very interesting to see whether such large performance gains are also sustained in NLP applications.

Finally, our results clearly demonstrate that transductive learning can lead to huge accuracy gains. Thus, it would be wasteful to use standard induction in settings where transduction is applicable.

# Acknowledgments

This research was partially supported by the Israel Science Foundation, grant No. 710/18.

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
