# TransBoost: Improving the Best ImageNet Performance using Deep Transduction Supplementary Material

**Omer Belhasin**[*]
Department of Computer Science
Technion - Israel Institute of Technology
omer.be@cs.technion.ac.il

**Guy Bar-Shalom**[*]
Department of Computer Science
Technion - Israel Institute of Technology
guy.b@cs.technion.ac.il

**Ran El-Yaniv**
Department of Computer Science
Technion - Israel Institute of Technology, Deci.AI
rani@cs.technion.ac.il

## A    TransBoost Applications

In general TransBoost is particularly useful when we are able to accumulate a test set of instances and then finetune a specialized model to predict their labels. This setting has numerous use cases in various application fields including:

**Medicine**    Medical diagnosis is one possible meaningful use case. In this case, medical records can be gathered on a daily or weekly basis. TransBoost can then be used to finetune transductive models on top of existing inductive models in order to provide more reliable results for these specific records.

**FinTech**    A possible use case is portfolio management (not high frequency trading), where the model should recommend stocks for trading based on weekly or monthly activities. In order to predict actions better than an inductive model, TransBoost can train specialized models based on up-to-date activities and then provide better trading recommendations. Another example is risk assessment for loans or insurances, where many financial companies are already using machine learning (inductive) models to predict the risk. When it comes to loans, financial records can be gathered and then a specialized model can be finetuned by TransBoost to assess the risk for particular clients. In this context, TransBoost can be used to lower insurance premiums based on client-specific profiles.

**Targeted Advertising**    Recommendation systems for marketing are another strong use case. Here, users' recent activities are used to recommend products in active campaigns. TransBoost can be applied to build a specialized model on a daily or weekly basis in order to recommend accurately user-specific products based on their activities.

**Homeland Security**    Hazard prediction for governments is another strong application. If, for instance, we need to predict whether a dangerous event will occur based on some relevant documents provided by open or closed sources. TransBoost can be used to provide this data-specific predictions. It is worth mentioning that training a general inductive model to predict this task is much more difficult.

**Data Analytics**    An example of a possible use case is the analysis of photos or videos in Google Photos (or similar systems). The photos can be accumulated by their location (for instance) and then a TransBoost model can be trained to produce a photo/video summary that is biased towards these specific photos (e.g. photos-specific effects).

---

[*]Equal contribution.

36th Conference on Neural Information Processing Systems (NeurIPS 2022).

## B  The Similarity Function is Bounded

In the following, we claim and prove that $\mathcal{S}_f$ is bounded.

**Claim 1.** *Let $f : \mathcal{X} \rightarrow \{1, \ldots C\}$ be a classification model and $x_1, x_2 \in \mathcal{X}$ input instances, where $C$ is the number of classes. Accordingly, our implementation of TransBoost's symmetric pairwise similarity function, $\mathcal{S}_f$ (4), satisfies the following:*

$$0 \leq \mathcal{S}_f(x_1, x_2) \leq \sqrt{2}.$$

*Proof.*

Recalling our implementation of $\mathcal{S}_f$ (4),

$$\mathcal{S}_f(x_1, x_2) \triangleq \sqrt{2} - \|\hat{p}(x_1|f) - \hat{p}(x_2|f)\|_2 \,,$$

where $\hat{p}(x|f)$ is the empirical class probabilities of $f$ applied on $x$. Since $\| \cdot \|$ is non-negative, we obtain $\mathcal{S}_f(x_1, x_2) \leq \sqrt{2}$. Now we need to show that $\mathcal{S}_f(x_1, x_2) \geq 0$, which holds iff, $\|\hat{p}(x_1|f) - \hat{p}(x_2|f)\|_2 \leq \sqrt{2}$, iff

$$\|\hat{p}(x_1|f) - \hat{p}(x_2|f)\|_2^2 \leq 2.$$

Considering the left-hand side of the inequality,

$$\|\hat{p}(x_1|f) - \hat{p}(x_2|f)\|_2^2 = \|\hat{p}(x_1|f)\|_2^2 + \|\hat{p}(x_2|f)\|_2^2 - 2\langle\hat{p}(x_1|f), \hat{p}(x_2|f)\rangle.$$

Following a sub case of Hölder inequality, $\forall v \in \mathbb{R}^n \; ; \; \|v\|_2 \leq \|v\|_1$. Therefore, using the last equation we obtain:

$$
\begin{aligned}
\|\hat{p}(x_1|f) - \hat{p}(x_2|f)\|_2^2 &= \|\hat{p}(x_1|f)\|_2^2 + \|\hat{p}(x_2|f)\|_2^2 - 2\langle\hat{p}(x_1|f), \hat{p}(x_2|f)\rangle \\
&\leq \|\hat{p}(x_1|f)\|_1^2 + \|\hat{p}(x_2|f)\|_1^2 - 2\langle\hat{p}(x_1|f), \hat{p}(x_2|f)\rangle \\
&= 2 - 2\langle\hat{p}(x_1|f), \hat{p}(x_2|f)\rangle && \text{(S.1)} \\
&\leq 2. && \text{(S.2)}
\end{aligned}
$$

Where (S.1, S.2) hold since $\hat{p}$ is a probability vector. Hence for any $x \in \mathcal{X}$, $\|\hat{p}(x|f)\|_1 = \sum_{i=1}^C |\hat{p}_i(x|f)| = 1$ and $\forall 1 \leq i \leq C \; ; \; 0 \leq \hat{p}_i(x|f) \leq 1$. ∎

## C  Transduction Training Details of SimCLR and SimCLRv2

The SimCLR [1] algorithm consists of self-supervised pretraining and supervised fine-tuning. Sim-CLRv2 [2] incorporates approximately the same steps, but adds a final knowledge distillation [3] stage that has been found powerful in inductive SSL (see [2] for further details).

SimCLR (and SimCLRv2), however, are designed for SSL rather than transduction; therfore, we will now elaborate how they were trained for transduction. As a first point, we note that both SimCLR and SimCLRv2 were applied to the ResNet50 and ResNet152 architectures using pretrained weights provided by the authors. Next, we fine-tuned the model, using both the labeled training set as well as the unlabeled test set, for another 100 epochs. We followed the pretraining scheme as suggested by the authors, except for the batch size which was set to 1024 due to lack of computational resources (they used 4096). In the second step of supervised fine-tuning, we followed the SimCLRv2 instructions given by the authors. For the final stage (self-distillation), SimCLRv2 optimizes a weighted loss function based on labeled training instances applied to standard CE and unlabeled training instances applied to the CE distillation loss. In our case, the unlabeled set is the transductive set.

## D  Supplement for the Additional Vision Datasets Experiments

Here we discuss the details in the experiments on TransBoost using the additional vision datasets that were introduced in Section 5.3, as well as an additional observation offered in Appendix D.2. Table S.1 compares the TransBoost performance to the standard inductive (fully supervised) performance and highlights some relevant properties of the datasets we used.

| Dataset | Train (+Validation) | Test | Classes | Reso-lution | Inductive / Transductive Top1 Acc. (%) | Improv. (%) |
|---|---|---|---|---|---|---|
| CIFAR-10 [4] | 50,000 | 10,000 | 10 | 32 | 97.60 / **97.61** | +0.01 |
| CIFAR-100 [4] | 50,000 | 10,000 | 100 | 32 | 82.77 / **85.29** | +2.52 |
| Stanford Cars [5] | 8,144 | 8,014 | 196 | 224 | 87.15 / **90.80** | +3.65 |
| Flowers-102 [6] | 7,169 | 1,020 | 102 | 224 | 94.86 / **97.85** | +2.99 |
| SUN-397 [7] | 87,003 | 21,750 | 397 | 224 | 93.25 / **95.94** | +2.69 |
| FGVC Aircraft [8] | 6,800 | 3,400 | 102 | 224 | 78.46 / **83.80** | +5.34 |
| DTD [9] | 3,760 | 1,880 | 47 | 224 | 74.79 / **76.49** | +1.70 |
| Food-101 [10] | 75,750 | 25,250 | 101 | 224 | 79.88 / **84.30** | +4.42 |

Table S.1: A study comparing TransBoost transductive performance to standard inductive (fully supervised) performance on various vision datasets (green) using ResNet50. TransBoost's best result is highlighted in yellow.

### D.1 Training Details on The Additional Vision Datasets

We now elaborate on the training details (inductive pre-training and transductive fine-tuning) of the additional vision datasets experiments. In all our experiments, we employed the ResNet50 architecture with the first max-pooling layer turned off.

In the case of non-CIFAR datasets, we started with an encoder model pre-trained on ImageNet, then fine-tuned it on each dataset, in conjunction with a newly initialized classifier. We chose to start with pre-trained weights on ImageNet, in order to achieve superior results on each dataset for later comparisons. The inductive training procedure on ImageNet was conducted using the SGD optimizer with a momentum of 0.9, weight decay of $10^{-4}$, and nesterov. We used a learning rate of $10^{-3}$ and a cosine schedule for 300 epochs with a batch size of 128, while in the first 90 epochs we applied a linear warmup. Finally, transductive fine-tuning was applied as detailed in Section 5.1, except for the batch size, which was set to 128 (rather than 1024).

With the CIFAR-10 and CIFAR-100 datasets, we started with a random initialization of ResNet50. The model was trained inductively using the SGD optimizer with a momentum of 0.9, weight decay of $10^{-4}$, and nesterov. We used a learning rate of 0.1 and a cosine schedule for 250 epochs with a batch size of 128, while in the first 75 epochs we applied a linear warmup. Transductive fine-tuning was performed using the SGD optimizer with a learning rate of $10^{-4}$ based on a validation set. Other details follow our standard training scheme, as described in Section 5.1, except for the batch size which was set to 128 (rather than 1024).

### D.2 Transductive Performance Gain Improves as the Number of Classes Grows

Table S.1 displays an interesting pattern regarding the number of classes. For experiments conducted using both non-CIFAR and CIFAR groups, we observe that TransBoost performs better on datasets with many classes than on datasets with a few classes. This pattern becomes blurrier as the number of classes increases. Specifically, among non-CIFAR experiments, it can be seen that the experiment on the DTD dataset attained the smallest improvement. This effect is also observed among the CIFAR group while the CIFAR-10 dataset was achieved the smallest accuracy gain. On the basis of these observations, we hypothesize that the more classes the dataset has, the more likely similar representations, which belong to different classes, will be found and, therefore, there are more target test instances to improve.

## E  Supplement for the ImageNet Subsets Experiments

This section provides a supplementary in-depth discussion of our experiments on ImageNet training and test subsets, which are varying in their sizes (introduced in Section 5.4). In this study, we used the ResNet50 architecture that was pretrained on ImageNet combinations of training and test subsets (using PyTorch's standard training recipe). For experiments that do not utilize the entire training set, we applied the TransBoost procedure with a batch size of 128. Although TransBoost was optimized for a specific test set, here we evaluate its performance in various inductive settings, where unseen test instances are examined at test time. Additionally, we present a comprehensive analysis

of TransBoost's performance in various transductive settings as well as a standard inductive (fully supervised) performance analysis as a baseline. The complete results are presented in Table S.2.

| ImageNet Fraction | | Instances Per Class | | Inductive Baseline | Transductive TransBoost | Inductive Baseline | Inductive TransBoost |
|---|---|---|---|---|---|---|---|
| Train | / | Test | | Top-1 (%) | / Improv. (%) | Top-1. (%) | / Improv (%) |
| 5% / | 64 | 10% / | 5 | 35.68 / **39.10** | +3.42 | **36.08** / 35.01 | -1.07 |
| | | 25% / | 13 | 36.03 / **38.92** | +3.18 | **36.04** / 35.49 | -0.55 |
| | | 50% / | 25 | 36.30 / **38.73** | +3.25 | 35.78 / **35.94** | +0.16 |
| | | 100% / | 50 | 36.04 / **38.01** | +1.97 | - | - |
| 10% / | 128 | 10% / | 5 | 50.70 / **53.04** | +2.34 | **50.27** / 48.80 | -1.47 |
| | | 25% / | 13 | 50.39 / **53.57** | +3.18 | **50.29** / 49.13 | -1.16 |
| | | 50% / | 25 | 50.63 / **53.88** | +3.25 | **50.00** / 49.78 | -0.22 |
| | | 100% / | 50 | 50.32 / **53.48** | +3.16 | - | - |
| 20% / | 256 | 10% / | 5 | 62.20 / **65.58** | +3.38 | **62.20** / 60.29 | -1.91 |
| | | 25% / | 13 | 61.78 / **65.78** | +4.00 | **62.35** / 60.63 | -1.72 |
| | | 50% / | 25 | 62.23 / **66.18** | +3.95 | **62.18** / 61.41 | -0.77 |
| | | 100% / | 50 | 62.20 / **66.07** | +3.87 | - | - |
| 100% / | ∼1300 | 10% / | 5 | 75.38 / **77.18** | +1.80 | **76.23** / 74.31 | -1.92 |
| | | 25% / | 13 | 75.53 / **77.87** | +2.34 | **76.36** / 74.46 | -1.90 |
| | | 50% / | 25 | 75.92 / **78.52** | +2.60 | **76.38** / 74.64 | -1.74 |
| | | 100% / | 50 | 76.15 / **79.03** | +2.88 | - | - |

Table S.2: A comprehensive analysis of TransBoost's performance on ImageNet subsets, both in transduction and induction, compared to the standard inductive (fully supervised) performance using ResNet50. The performance of TransBoost in transduction is highlighted in green and the performance of TransBoost in induction is highlighted in blue.

Table S.2 presents 16 experiments of TransBoost's procedure performed on all combinations of the ImageNet training set fractions: 5%, 10%, 20%, 100%; and the ImageNet test set fractions: 10%, 25%, 50%, 100%. The table's sections represent training set fractions, and the rows in each section represent test set fractions. There are two main column groups highlighted in green and blue. Green columns present comprehensive results in transductive settings, whereas blue columns present comprehensive results in inductive settings. Across the two settings (transductive / inductive), TransBoost's performance is compared to the standard inductive (fully supervised) performance. For instance, using 20% of the training set and 25% of the test set, we obtained the best top-1 accuracy gain of +4.00% in the transductive setting while the performance in the inductive setting degraded by -1.72%. We note that the performance in the inductive setting (highlighted in blue) is evaluated using the test instances that were not used at training time.

As can be seen, TransBoost consistently outperforms the baselines in transductive settings while the baselines almost consistently outperform TransBoost in inductive settings. This behavior is indeed what one may expect from real transductive models, which will only be asked for the given test set. Interestingly, TransBoost's best top-1 accuracy gain – an impressive improvement of +4.00% – was achieved by the experiment that used 20% of the training set and 25% of the test set. Our best top-1 accuracy (79.03%) was attained in the experiment that used the entire training and test sets. For further details, we refer the reader to Appendix E.1 and Appendix E.2, below.

### E.1 TransBoost's Inductive Performance Increases with Test Set Size

In Figure S.1, we present an interesting trend of TransBoost's top-1 accuracy gains in various inductive settings (derived from Table S.2). Note that the fractions described in the figure legend refer to the test instances that participated in the training procedure, while the inductive performance was examined using the rest of the instances. We have already discussed the fact that almost all inductive baselines outperform TransBoost in all inductive settings. This can be expected from a transductive model that specializes in a particular test set. Figure S.1, however, reveals an interesting pattern: the more test instances are provided at training time, the better TransBoost's performance in the inductive

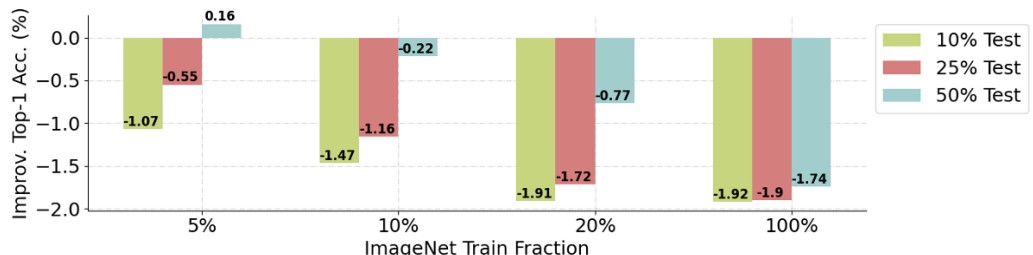

Figure S.1: TransBoost's inductive performance using ResNet50 on ImageNet training and test subsets.

setting. This observation implies that our proposed optimization procedure learns better to generalize unseen instances when the ratio of test instances (out of the training and test instances combined) is significant. Moreover, it is apparent that performance deteriorates with increasing training sets.

## E.2 TransBoost's Loss is Consistently Improved

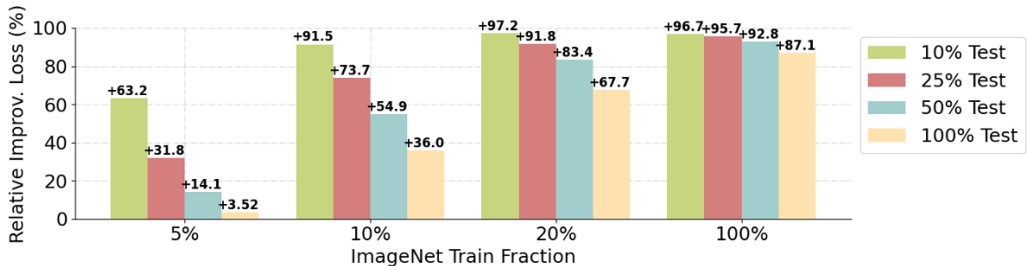

Figure S.2: Relative improvements in our proposed loss component, $\mathcal{L}_{\text{TransBoost}}$, using ResNet50 on ImageNet training and test subsets.

Here we provide an additional analysis of the relative improvement in our proposed loss component, $\mathcal{L}_{\text{TransBoost}}$ (see Section 4.1), across the experiments that we described above. Specifically, we approximated TransBoost's loss following Algorithm 1 (the approximation stage only) on the inductive baselines as well as on TransBoost's output models in the transductive setting. Then, we calculated their relative improvements in (%) and presented the results in Figure S.2. For example, our best relative improvement in $\mathcal{L}_{\text{TransBoost}}$ was given by the experiment that used 20% of the training set and 10% of the test set with a relative improvement of +97.2%.

As shown in Figure S.2, the larger the test set that TransBoost uses, the smaller its improvement on our loss component $\mathcal{L}_{\text{TransBoost}}$. Accordingly, larger test sets may be harder to optimize than smaller ones, and may require more optimization steps. The larger the training set, the greater the improvement on the $\mathcal{L}_{\text{TransBoost}}$. This suggests that the greater the prior knowledge of the model is, the easier it is to optimize $\mathcal{L}_{\text{TransBoost}}$.