# OpenReview forum: "TransBoost: Improving the Best ImageNet Performance using Deep Transduction"
_NeurIPS.cc/2022/Conference — NeurIPS 2022 Accept_

### Official Review · Reviewer_GxTe · 2022-07-11

**Rating:** 8
**Confidence:** 3
**Soundness:** 4 excellent
**Presentation:** 4 excellent
**Contribution:** 3 good

**Summary:**

Paper proposes TransBoost, a transductive learning loss function that improves performance in transductive image classification in domains where large datasets and test sets are available. The work draws inspiration from TSVM and proposed a transductive loss regularization term that discriminates unlabelled examples that have similar class probabilities but shouldn't. Extensive experiments demonstrate that TransBoost improves performance on various architectures, datasets, and settings. Ablation studies provide interesting insights into the behaviour of TransBoost under different conditions.

**Questions:**

Suggestions are included in the above sections. An overall strong submission that could be further strengthened by additional analysis of TransBoosts uneven behaviour with respect to architectures.

**Limitations:**

Yes, the authors provide an in-depth analysis of the technical limitations of the work and provide directions for future research. There is, however, no explicit discussion of societal impacts. I believe the work could benefit from a concise analysis of its societal impact and if any could potentially be negative.

**Strengths And Weaknesses:**

Strengths:
- Paper is incredibly well-written and very easy to follow.
- TransBoost is an interesting and relatively novel idea. It effectively bridges the gap in performance from t-FSL methods that have been widely developed. The algorithmic choices are sound and well-motivated.
- Experiments are extensive and provide sufficient evidence to establish state-of-the-art performance.

(Small )Weaknesses:
- Certain results suggest that TransBoost is more applicable to some architectures than others. Although performance improvements are universal, some methods improve much more, and some even surpass their otherwise more inductively powerful counterparts. This suggests that TransBoost is algorithmically better suited for certain architectures, but the reasoning is primarily speculated. It would be useful to see further analysis as to why for instance, Transformer based architectures benefit from TransBoost?
- One would presume that alternatively modifying the loss function to group together similar samples would benefit performance, albeit not to the same extent. However, the observation that it degrades performance may suggest that transductive learning can produce fragile optimization manifolds that should be carefully designed. Further elaboration on this front could be useful.

---

> ### Author Response · Authors · 2022-07-30
> **Future work**
>
> Thank you for your positive review! We appreciate your feedback and are very pleased that our work inspires you. As well, we're curious about your suggestions, like whether TransBoost is more applicable to some architectures than others and what rules TransBoost follows from a theoretical perspective in order to understand how to maximize its performance. We'll leave these questions for future work, as we also mentioned in Section 6 (Concluding Remarks) for some related research questions.

---

> > ### Comment · Reviewer_GxTe · 2022-08-09
> > **Maintaining recommendation for acceptance.**
> >
> > After considering comments by other reviewers, authors, and rebuttals, I believe that the submission has demonstrated enough merit for acceptance. I also believe that the community would benefit from the work.

---

### Official Review · Reviewer_XZ2m · 2022-07-13

**Rating:** 5
**Confidence:** 4
**Soundness:** 2 fair
**Presentation:** 3 good
**Contribution:** 2 fair

**Summary:**

This paper investigates deep transductive learning where both a labeled training sample and an unlabeled test sample are provided at training time. The goal of deep transductive learning is to achieve high accuracy on the test set. To this end, the authors propose TransBoost as a procedure to fine-tune any deep neural model using a labeled training sample and an unlabeled test sample. Extensive experiments are conducted to verify the effectiveness of the proposed method.


**Questions:**

Please see details in Cons.  I would like to suggest the authors define a meaningful task. Eqn.(2) borrows ideas from TSVM and it would have a positive effect in test time training [1]

[1] Sun et al. Test-Time Training with Self-Supervision for Generalization under Distribution Shifts. ICML 2020.


**Limitations:**

Yes

**Strengths And Weaknesses:**

**Pros:**
1. The paper is well-written and well-organized.
2. Investigating conventional statistical machine learning in the context of deep learning is interesting and meaningful.
3. Extensive experiments are conducted to verify the effectiveness of the proposed task and method.

**Cons:**
1. The task of deep transductive learning has a severe issue of overfitting target test set. Although Transboost does not use the true label of the test set, it implicitly uses pseudo labels, i.e. similarity of image pairs. In this sense, the performance would boost a lot in any deep models as shown in Fig.1.

---

> ### Author Response · Authors · 2022-07-30
> **Providing support for TransBoost motivations**
>
> Thank you for your review and your suggestion. We will do our best to explain in detail why transductive learning is a meaningful task and detail various applications for TransBoost.
>
> First, let's briefly discuss  transductive learning. Transduction was first introduced by Vapnik [1] in the context of statistical learning theory. In relation to classical machine learning, it has been extensively examined in the literature, including the empirical work [2] that applied TSVM to various datasets of text classification to demonstrate accuracy gains compared to standard SVM. The work in [2] first demonstrated that a transductive algorithm could boost the performance of an inductive models (not neural models). As far as deep learning is concerned, transduction has only been briefly addressed. Therefore, we developed TransBoost to take advantage of transduction in deep learning, and we demonstrate overwhelming and consistent improvements (relative to standard inductive deep learning) that should be exploited whenever transductive learning is applicable.
>
> [1] Vladimir Vapnik. The nature of statistical learning theory. Springer science \& business media, 1999.
>
> [2] Thorsten Joachims et al. Transductive inference for text classification using support vector machines. In Icml, volume 99, pages 200–209, 1999.
>
> *"The task of deep transductive learning has a severe issue of overfitting target test set"*
>
> Sorry for the misunderstanding. Overfitting is perhaps the wrong word here and the correct one is probably biasing. Indeed biasing the model towards the given (unlabeled) test set is precisely what transductive learning is about.
>  In the transductive setting we are given both a labeled and an unlabeled set at training time and the goal is to predict only the labels of the given unlabeled set (the test set) as accurately as possible, without caring about any unseen instances. In section 2 (Problem Formulation), we state that our target objective is only the loss on the given test set (Equation 1). Perhaps this is what ignited your concern about overfitiing/biasing. However, the legitimate biasing in transduction enables improved performance.
>
> *"Although Transboost does not use the true label of the test set, it implicitly uses pseudo labels, i.e. similarity of image pairs. In this sense, the performance would boost a lot in any deep models as shown in Fig.1."*
>
> First we should all agree that using pseudo labels is a perfectly valid ingredient.
> However, observe that simply using pseudo labels is not enough in order to achieve the accuracy gains that we presented in Figure 1.
> First, we have preliminary results on ResNet architectures that show that simply training each model with both labeled data and unlabeled data with its pseudo labels via the standard cross entropy does not provide any accuracy gain (the performance remains the same). Much of the power of transductive learning is achieved through analyzing the given test set as a group. Thus, we designed TransBoost to take advantage of the test set as a group instead of one instance at a time as in inductive settings. In Section 5.4 we empirically demonstrated this idea. We observed that whenever TransBoost uses the entire training set, its performance increases as the test set size grows.
>
> *"I would like to suggest the authors define a meaningful task. Eqn.(2) borrows ideas from TSVM and it would have a positive effect in test time training"*
>
> We appreciate your suggestion. The Test-Time Training approach is indeed an interesting application for TransBoost to improve performance under distribution shifts. We'll add a discussion about it in Section 6 (Concluding Remarks) in the final version of the paper. Thank you for the good comment.
> Meanwhile, An appendix section with detailed elaborations on TransBoost applications has been added to a revised paper and prioritized as Appendix A. In general TransBoost is particularly useful when we are able to accumulate a test set of instances and then finetune a specialized model to predict their labels, as we demonstrated in our paper. There are numerous applications for this setting. Our paper briefly mentioned medical diagnosis (Introduction, Paragraph 3) and now in Appendix A we elaborate on additional applications in various fields: medicine, fintech, targeted advertising, homeland security, and data analytics (see Appendix A for more information). Using TransBoost, a user can achieve more accurate predictions in each task while using an efficient and simple framework on top of existing inductive (standard) models.

---

> > ### Comment · Reviewer_XZ2m · 2022-08-09
> > **Response to rebuttal**
> >
> > Thank you for your detailed rebuttal. Although the authors have clarified the concept of overfitting and biasing,  I still have doubt whether it is meaningful to use test data for image-pair learning. In my view, the performance can be definitely (with appropriate usage) improved when test data is used in training stage. Note that transduction learning typically have source domain and target domain, which is often the case in practice. However, test set is often used to test the performance of a model and does not exist in true scenario. I decide to keep my rating unchanged.

---

> > > ### Author Response · Authors · 2022-08-09
> > > **Please read about transductive learning**
> > >
> > > Unfortunately (to us) your view and beliefs on transductive learning are wrong.
> > > It is recommended that you read the Wikipedia entry:
> > > https://en.wikipedia.org/wiki/Transduction_(machine_learning).
> > > This page provides some basic but detailed information on transductive learning, as well as an example problem to illustrate some of the unique properties of transduction against induction.

---

> > > > ### Comment · Reviewer_GxTe · 2022-08-09
> > > > **Let's be fair to the authors.**
> > > >
> > > > While I generally avoid directly weighing on other reviewers' comments, I believe that in this case, it's necessary. Transductive learning is a well-established problem domain, with extensive prior literature, especially in the field of few-shot learning. Furthermore, I am personally aware of industrial settings where transductive learning would be beneficial (such as the case where large unlabelled sets of images are available and we need to produce labels on all instances but want to label only a proportion of them). The authors have also clearly discussed bias issues that may arise.
> > > >
> > > > Therefore, I find it unfair to completely reject the submission on the basis of the problem domain not being convincing enough. I would be happy to discuss the specific merits of the method, the experiments, the limitations, and other factors in generating my final reading. However, the problem domain concerns do not provide grounds for rejecting the paper in my opinion.

---

> > > > ### Comment · Reviewer_XZ2m · 2022-08-10
> > > > **Thanks for the correction**
> > > >
> > > > After reading relevant materials, I realize that the unlabelled test data are usually used in transduction learning. The experimental results in Table 1 also shows that TransBoost outperforms self-supervised learning methods such as SimCLRv2 in transductive setting, which demonstrate its superiority. The practical value of transduction learning is clear in few-shot learning where only a few labeled samples are given. However, the benchmarks used in this paper, such as ImageNet, CIFAR10, el al., have a large number of labelled training data, it is unnecessary for us to use (unlabelled) test data. Anyway, this paper demonstrates the effectiveness of transduction learning in classification, even better than SSL methods. Hence, I would like to raise my rating to 5.

---

### Official Review · Reviewer_Sp8K · 2022-07-16

**Rating:** 5
**Confidence:** 3
**Soundness:** 2 fair
**Presentation:** 2 fair
**Contribution:** 2 fair

**Summary:**

The paper proposes TransBoost to deal with deep transductive learning. The method follows the large margin principle and includes the similarity and confidence of two instances in the loss function. The effectiveness of this method is examined by using a variety of pre-trained neural networks, datasets, and baseline models. It shows good performance compared to inductive classification as well as versus SSL and t-FSL methods

**Questions:**

1. Some paragraphs are hard to read. For example, in the 3rd paragraph in Section 4, what does it mean: we incentivize test sample pairs that are likely to be different in their classes to also be different in their empirical class probabilities while preserving the prior knowledge of $f_{\theta}$?
2. The toy example in Section 4.2 is unclear:
(a) How is the p(x|f) calculated with SVM?
(b) Although TransBoost loss is lower in T-SVM, does the decision boundary of T-SVM provides better separation of the test instances? So what is the distribution of test instances?
3 Typo?
(a) Section 5.5: separating the different -> separating the difference
(b) Figure 2: ... example indicating ... -> ... example indicates ...

**Strengths And Weaknesses:**

Strengths:
It provides extensive experiments on multiple neural network architectures and datasets and shows good improvement.

Weakness:
The paper is not in the submission format, there is no line number.
Clarity: some paragraphs are hard to read.
Significance: the author doesn't provide a strong motivation for why their work is important

---

> ### Author Response · Authors · 2022-07-30
> **Providing support for TransBoost motivations and in-depth additional explanations**
>
> Thank you for your review and your suggestions. We are committed to address all the questions and concerns raised in your review.
>
> *"The paper is not in the submission format, there is no line number."*
>
> Thanks, you are correct, we have added the line numbers and uploaded a revised paper.
>
> *"Significance: the author doesn't provide a strong motivation for why their work is important"*
>
> We agree with this concern and thank you for this good comment. Indeed, we have not sufficiently motivated deep transductive learning and haven't elaborated on useful applications for TransBoost. An appendix section with detailed elaborations on TransBoost applications has been added to a revised paper and prioritized as Appendix A.
> In general TransBoost is particularly useful when we are able to accumulate a test set of instances and then finetune a specialized model to predict their labels, as we demonstrated in our paper. There are numerous applications for this setting. Our paper briefly mentioned medical diagnosis (Introduction, Paragraph 3) and now in Appendix A we elaborate on additional applications in various fields: medicine, fintech, targeted advertising, homeland security, and data analytics (see Appendix A for more information). Using TransBoost, a user can achieve more accurate predictions in each task while using an efficient and simple framework on top of existing inductive (standard) models.
>
> *"in the 3rd paragraph in Section 4, what does it mean: we incentivize test sample pairs that are likely to be different in their classes to also be different in their empirical class probabilities while preserving the prior knowledge of $f_\theta$?"*
>
> The following statement was made in this section to give an overview of TransBoost's process after it was introduced formally. Let us elaborate on this intuition.
> TransBoost loss in Equation 2 consists of three components - a confidence function $\kappa$, a selection function $\delta$, and a similarity function $\mathcal{S}$. Informally, it was designed to distinguish representations of pairs that are likely ($\kappa$) to belong to different classes ($\delta$), but appear to be similar ($\mathcal{S}$). For example, if we have a pair of unlabeled test samples where one is predicted to be "Dog", and the other is predicted to be "Cat", TransBoost will calculate its loss based on their similarity in their class probability vectors, so that when Dog and Cat have similar class probability vectors (as defined by Equation 4), they will get a higher loss than when Dog and Cat have dissimilar class probability vectors. As an intuition, we wrote: "preserving the prior knowledge of $f_\theta$" to emphasize that we are given a pretrained inductive model that performs meaningfully on the test set, so our method preserves the accuracy of the given model on the labeled training set while boosting the performance on the test set.
> Anyway, thank you for your note, we will fix this in the revised paper.
>
> *"(a) How is the $p(x|f)$ calculated with SVM?"*
>
> We agree to add an appendix section that includes these technical details. Given a test point $x\in\mathbb{R}^2$ we calculated its binary class probability vector $p(x|f)$ via  standard logistic regression. Formally, Let $w\in\mathbb{R}^2$ and $b\in\mathbb{R}$ denote the SVM's coefficients and bias term after training. We first calculated its linear score $z=w^Tx+b$. Then, we applied the logistic function $f(z)=\frac{1}{1+\exp(-z)}$ to calculate the probability score for the instance belonging to the first class (the probability score for the instance belonging to the second class is $1-f(z)$).
>
> *"(b) Although TransBoost loss is lower in T-SVM, does the decision boundary of T-SVM provides better separation of the test instances? So what is the distribution of test instances?"*
>
> The purpose of this toy example is to quantitatively demonstrate why training with the TransBoost loss encourages large margins on a test set (similar to TSVM). In this example, we synthesized the labeled training points and unlabeled test points without considering their real distribution (because it's synthetic). Therefore, this example does not claim anything about the model's performance when a large margin principle is taken into account on the test set, but only illustrates that TransBoost encourages this behavior.
> In classical machine learning, large margin methods were extensively researched. A relevant example may be the work [1] that demonstrated impressive accuracy gains when TSVM was compared with SVM on several binary text classification datasets. We believe this work can answer your question on how large margins can help the test set perform better (Figure 5 in [1]).
>
> [1] Thorsten Joachims et al. Transductive inference for text classification using support vector machines. In Icml, volume 99, pages 200–209, 1999.
>
> Thank you for the other minor comments, we fixed them and uploaded a revised version.

---

### Meta-Review · Area_Chair_tL19 · 2022-08-25

**Recommendation:** Accept
**Confidence:** Certain

**Metareview:**

This paper initially received mixed opinions. After intensive author-reviewer and reviewer-reviewer discussions, all reviewers converged and recommended acceptance. AC recommends accepting the paper.

**Award:**

No

---

### Decision · Program_Chairs · 2022-09-14

Accept